# Fractional-Order Electrical Modeling of Aluminum Coated via Plasma Electro-Oxidation and Thermal Spray Methods to Optimize Radiofrequency Medical Devices

**DOI:** 10.3390/s24082563

**Published:** 2024-04-17

**Authors:** Noelia Vaquero-Gallardo, Oliver Millán-Blasco, Herminio Martínez-García

**Affiliations:** Department of Electronics Engineering, Eastern Barcelona School of Engineering (EEBE), Technical University of Catalonia—BarcelonaTech (UPC), E-08019 Barcelona, Spain; noelia.vaquero@upc.edu

**Keywords:** vector network analyzer (VNA), electrical equivalent model, fractional order, surface coating techniques, plasma electro-oxidation (PEO), thermal spray (TS), radiofrequency medical devices

## Abstract

Active medical devices rely on a source of energy that is applied to the human body for specific purposes such as electrosurgery, ultrasounds for breaking up kidney stones (lithotripsy), laser irradiation, and other medical techniques and procedures that are extensively used. These systems must provide adequate working power with a commitment not to produce side effects on patients. Therefore, the materials used in these devices must effectively transmit energy, allow for security control, sense real-time variations in case of any issues, and ensure the implementation of closed-loop systems for control. This work extends to the experimental data adjustment of some different coating techniques based on plasma electro-oxidation (PEO) and thermal spray (TS) using fractional-order models. According to the physical structure of the coating in different coating techniques, Cole family models were selected. The experimental data were obtained by means of a vector network analyzer (VNA) in the frequency spectrum from 0.3 MHz to 5 MHz. The results show that some models from the Cole family (the single-dispersion model and inductive model) offered a goodness of fit to the experimental impedance in terms of RMSE error and a squared error R^2^ close to unity. The use of this type of fractional-order electrical model allows an adjustment with a very small number of elements compared to integer-order models, facilitating its use and a consequent reduction in instrumentation cost and the development of control devices that are more robust and easily miniaturized for embedded applications. Additionally, fractional-order models allow for more accurate assessment in industrial and medical applications.

## 1. Introduction

Current trends have become increasingly focused on the search for safer, faster, and more accurate practices for energy transfer in multidisciplinary sectors such as industry, domestic, medicine and telecommunications. These practices involve applying a controlled energy source above the tissue or another system to induce a specific response or intrinsic changes in its normal behavior. For example, this could include temperature variation for welding in industrial applications, tissue overheating for ablation in electrosurgery or magnetic resonance imaging [1,2,3,4,5].

In this work, a system based on emitting a radiofrequency signal to promote tissue ablation is under material characterization. Figure 1 shows a schematic representation of the system, which integrates various key components, namely, surface coating, energy source, tissue and the current signal direction.

The treatment consists of supplying a radiofrequency (RF) signal of 0.3 MHz to 5 MHz over the skin tissue. The passage of that signal contributes to an increase in temperature from 55 to 80 °C on the tissue and the destruction of it [5,6,7]. The element to transmit the energy comprises signal energy transmitters and the tissue.

The application of this energy must be carefully applied to the tissue to induce controlled heating in the layer of the skin for cutting [1,2,3,5,8]. It is crucial to ensure precise delivery control of this energy to prevent any aftereffects [9,10,11,12]. Therefore, an appropriate sensing system must be implemented, considering the geometry and materials of the system, to monitor and detect variables in real time, enabling accurate closed-loop control [13,14,15,16]. Furthermore, the components of the structure are constructed from aluminum, specifically type 6063 (0.7 Mg–0.4 Si), which are directly in contact with the transmitters of radiofrequency signals. This metal is light and is widely used in the industry due to its strength-to-weight ratio, low density, ease of manufacturing, low cost, and high electrical and thermal conductivity [17,18,19,20]. Consequently, an inefficient treatment could be applied by virtue of the fact that the system structure is more electrically conductive than the tissue, not allowing the passage of the radiofrequency through it. These requirements and limitations necessitate the application of a surface coating modification on the components to electrically isolate the material without reducing the thermal conductivity, ensuring precise application.

This work is extended to the experimental data adjustment of some different coating techniques on refrigeration parts based on plasma electro-oxidation (PEO) and thermal spray (TS) using fractional-order models. These models were proposed due to the potential reduction in required elements compared to electrical equivalent models using an integer-order approach, implying a significant advantage in terms of low-cost instrumentation [21,22,23,24].

Moreover, a more robust and easily miniaturized real-time sensing embedded system could be implemented. Furthermore, a more efficient adjustment of energy utilization could be achieved through proper implementation [9,10,25,26].

The most suitable surface coating for the application of this work was determined by comparing the electrode impedance and the tissue impedance to achieve the highest energy transfer. Consequently, in future lines of that research work, the aim is to characterize the tissue with fractional-order electrical models and, subsequently, to compare the experimental results obtained in that work with the further electrical equivalent models of the tissue. The fractional-order approach implies significant benefits for advancing goals in industrial and medical research, warranting further interesting research [26,27,28].

### Surface Coating Techniques

Surface coating is an engineering process in which the surface of a material is subjected to another overlay, such as powder, film or bulk for intended applications based on Matthiessen’s rule [29,30]. Coating methods are available in a wide variety due to the enormous diversity of applications [17]. The most versatile and well-known surface coating method in industrial applications is anodizing. This process forms a thin oxide film on the metal and confers enhanced electrical, corrosion and wear resistances, along with the possibility of modifying the color and texture of the surface [31].

The oxidation method of coating the surface of the metal is the method that enhances the characteristics of the outer layer of the metal [32]. The outcome is an alumina layer (Al_2_O_3_) that exhibits increased hardness, chemical resistance, good mechanical strength, transparency, and good insulating properties [33].

In that case, plasma electrode oxidation (PEO) and thermal spray (TS) have been selected as oxidation coating methods due to their isolation properties [34,35,36,37] and biocompatibility evaluation [38,39,40,41,42].

The first method, PEO, is a type of anodizing method where oxide coating is formed through micro-arc providing a strong porous oxide layer with a uniform structure compared to other anodizing family methods. According to PEO conditions as substrate, applied current density, duty cycle cell voltage, electrolyte, temperature, and duration have a huge effect on oxide layer features [43]. The second method, TS, is another oxidation method based on the melting of a metal wire in a flame generated by the combustion of fuel with oxygen, electric energy, or cold spraying melted metal. Afterward, the surface was atomized by compressed gas, propelling metal droplets onto a substrate to build up a coating.

The paper is organized as follows: Section 2 provides an overview of the distinct 6063 aluminum treated with the selected surface coatings. Meanwhile, the measurement system and the procedure for obtaining thermos-electrical data are covered. Results and discussion are presented in Section 3. Finally, the concluding remarks of this work are summarized in Section 4.

## 2. Materials and Methods

Both coatings methods were applied to 50 mm × 50 mm × 7 mm samples of aluminum type 6063. A coating of 30 mm × 50 mm has been applied on a single face of each sample.

In total, seven coating samples were analyzed, each processed in different laboratories. Five samples with different thicknesses were coated by means of PEO and three samples with different thicknesses were coated by means of Thermal Spray. Two samples of that group also include ceramic paint (CP) in combination with TS. Figure 2 presents the samples, and their configurations are detailed in Table 1.

As these methods are self-developed in two technology centers, there is no access to the implemented technological process. Therefore, the description provided is approximate, based on public indications from these laboratories and the literature. In the treatment of TS based on thermal spraying, it is claimed that technologies primarily based on high-velocity oxygen fuel combustion (HVOF-HVOAF) and detonation gun (D-gun) are utilized, while technologies based on electrical or cold methodologies are not typically employed [44].

In the case of PEO treatment, the equipment typically utilized has a bath capacity of 50 liters, with a maximum power of 20 kW at 50 A. Additionally, oxidation grows at a rate of up to 1 mm per minute and is conducted in three different phases, subjected to 415 V, 50 Hz and 64 A. Operational conditions for the PEO approach are highlighted in Table 2.

The methodology is described in three distinguishable parts: (A) measurement system, (B) electrical equivalent models, (C) fitting data processing. To summarize the complete procedure, some diagrams are provided down below in Section 2.

### 2.1. Measurement System

The study of the passive electrical behavior of biomaterials across a wide frequency range is conducted using various automated instruments. In this work, low-cost, professional-grade 6 GHz and 8.5 GHz PicoVNA^®^ Vector Network Analyzers (VNA) (Pico Technology, Cambridge, UK) and open-source software were selected to acquire the data. These instruments are based on time domain reflectometry and a non-destructive measurement method [45,46]. They calculate the impedance by analyzing the scattering (or S parameters) due to transmitted and reflected energy across the device under test (DUT).

The measurement configuration is composed of the impedance analyzer connected to the coated sample over a plate capable of providing specific temperatures. Through a coaxial cable, the signal is transmitted via the emitter electrode, which is in direct contact with the coating under evaluation. The coaxial cable shielding is connected to the non-coated surface under evaluation, running parallel to the emitter electrode. Figure 3 illustrates the experimental data acquisition process with the required subsystems.

### 2.2. Electrical Equivalent Models

An electrical equivalent is a circuit that exhibits identical electrical behavior to the original when studied from predefined terminals [46,47].

The objective is to construct an optimal circuit model that possesses physical significance and minimizes the number of variables.

The models are the combination of series and parallel two-element associations of integer and fractional order in order to give the best agreement between experimental and simulated spectra [48]. Fractional calculus has recently gained prominence across various science and engineering disciplines because they are more accurate than integer-order ones due to the extra degree of freedom in the fractional-order model than the conventional ones [21,48,49,50].

These measurements are adjusted using different fractional-order circuit models. From the literature, equivalent electrical models that fit correctly and that present a frequency behavior like the worked samples, according to the physical structure of the coatings have been selected. Specifically, these selected models are derived from the Cole family of models with fractional order [51,52]. The fractional factor is described by CPE (constant phase element) featuring two components, CPE-T and CPE-P, where the first element refers to pseudo capacitance and the second element is the factor related to the capacitance behavior. Notably, when the values of the constant phase elements (CPE-P) approach approximately 1, the CPE theoretically behaves like a capacitor [53].

These selected models are: (a) single-dispersion Cole model (a high-frequency resistor *R*∞, a resistor *R*1 and a constant phase element (CPE)), (b) double-dispersion Cole model (an additional parallel combination of a resistor (*R*2) and CPE in series with the single-dispersion Cole model), (c) capacitive model (CPE *Cα* and *R*1 and a constant phase element (CPE *Cβ*)), and (d) inductive model (a high-frequency resistor *R*∞, containing for the first time a fractional-order inductor *Lα* in addition to the CPE *Cβ*). Figure 4 shows the impedance models belonging to the Cole family.

### 2.3. Fitting Data Processing

The equivalent electrical circuit must generate spectra with appropriately chosen element values, which deviate minimally from the experimental results within a defined tolerance. All the elements of the proposed circuit must have a clear physical meaning [54]. Adjustment data for a specific model is an iterative process that ends when the discrepancy between observed values and expected values based on the concrete model has an acceptable level of goodness.

Data was analyzed and modeled using ZVIEW™ (v. es 4.0h), an equivalent circuit fitting software analysis tool by means of a Kramers–Kronig (KK) transform. The complex fit was adjusted for both the real and the imaginary parts. As the frequency spectrum is wide, the impedance was experimentally observed to vary over the range. Therefore, the number of iterations was adjusted until the chi-squared error was below 10^−5^. Ultimately, 25 average iterations were necessary for all the models to converge. Initially, resistive elements were fixed for each model as initial approximations, while the remaining elements were left free during the adjustment process.

The goodness of fit was evaluated using two key metrics: RMSE (root mean squared error) and R^2^ [55,56].

The RMSE measures the discrepancy between observed and predicted values in absolute terms, providing insight into the accuracy of the regression model. On the other hand, R-squared quantifies how well the linear regression model explains the variation in the dataset. Additionally, to assess the linearity of the model fit, the KK test was employed. This test compares the simulated Nyquist and Bode spectra with experimental data. Significant deviations between them may indicate a deficiency in adherence to the principles outlined above.

Afterward, electrical parameters for each coating type were analyzed and fitted. For each sample, the final measurements are derived from the average of five resulting iterations. Electrical equivalent fitting was conducted on four different models for each surface coating to validate the most accurate fitting model according to values obtained by the goodness of fit.

## 3. Results and Discussion

This section is divided into four parts. Initially, thermo-electrical parameters are presented to perform an analysis of their influence on electrical behavior in (1) input data for fitting, (2) electrical equivalent fitting, and (3) goodness of fit evaluation.

### 3.1. Input Data for Fitting

Bode and Nyquist plots for each coating are generated using a frequency sweep range. The results obtained according to impedance measurement for PEO coatings and TS coatings are presented below in Figure 5.

Analysis of the Bode plots in Figure 5 reveals that all sample coatings exhibit a first-order magnitude and capacitive phase behavior, where the Cole family is used for adjustment.

### 3.2. Electrical Equivalent Fitting

Nyquist plots are fitted for each coating. The results obtained are presented below in Figure 6 and Figure 7.

To avoid extending the graphical results, only the most significant results of the adjusted data are presented. Results for unadjusted data are not included due to the lack of fit in both capacitive and inductive models. This conclusion is supported by the goodness results presented in Figure 8c,d.

Results for the single-dispersion model are depicted in Figure 6. These results present an acceptable adjustment between the experimental data and the model in terms of the resulting Nyquist plot fitted and KK condition. Furthermore, Table 3 provides the numerical values of the equivalent circuit’s elements. As can be seen in Table 3, CPE1-P values are 1, which means the CPE element behaves like a capacitor at all temperature ranges.

For the inductive model, the results are presented in Figure 7. From the results in this figure, the inductive model presents an acceptable adjustment between the experimental data and the model in terms of the resulting Nyquist plot fitted and KK condition. The adjusted parameters obtained are summarized in Table 4.

### 3.3. Goodness of Fit Evaluation

RMSE and R^2^ values are plotted for each coating and model. The results obtained are presented below in Figure 8. Analyzing those models where the Nyquist representations fit the input data, it is observed that the RMSE errors are low, while the R^2^ errors are close to unity. This can be seen in Figure 8a,d for the single dispersion and inductive models.

In cases which have not obtained adequate fits observed in the Nyquist representations, the RMSE errors are very high compared to the correctly fitted ones (about 100 times more) and the R^2^ errors are far from unity. This can be seen in Figure 8b,c for the double dispersion and capacitive models, respectively.

The physical interpretation of the values corresponds to the microstructure of the samples, wherein the porosity of the outermost layer affects the deposition of the coating treatment. Consequently, it has a higher metallic presence, which implies a single time constant and allows the metal-coating interface to be considered as a resistor.

The next layer (inner layer) is characterized by a low porosity and is dominated by a higher coating content. Therefore, the CPE parameter can be modeled in parallel with the ionic resistance R. Outer and inner layers are represented in Figure 9 below.

## 4. Conclusions

This work is extended to the characterization of the electrical response of various coating techniques such as PEO and TS against a radiofrequency signal from 0.3 MHz to 5 MHz by fractional-order models. The extra degrees of freedom of that equivalent have enhanced the fitting parameters showing better accuracy. The single-dispersion model (an inductive model based on the Cole family models) has emerged as the best fit among different fractional models based on RMSE, R^2^ and KK condition goodness. The application of fractional orders enabled the representation of the electrical behavior of coatings with lower electrical elements compared to non-fractional orders. Therefore, all coatings are appropriate for treatment requirements but due to manufacturing and cost terms, those thicknesses low enough to ensure repeatability between samples in production are chosen. These results are pending for future lines of investigation where the tissue will be characterized in terms of fractional orders, and both results could be compared. With this second work, the surface coating could be finally selected according to impedance response behavior from coatings and tissue. The most suitable surface coating is one that presents a higher impedance response compared to other systems, thereby enhancing the transfer of energy, and improving the efficiency of application.

The effort to simulate these coatings using models that collect the magnitude response underlines the necessity of expanding circuit simulation tools to include fractional-order elements to simplify working in a future research line in terms of reduction of the required elements, thereby affording low-cost instrumentation. Additionally, it would facilitate the implementation of embedded sensing systems into these materials for real-time control and monitoring, resulting in a more robust closed-loop control system. According to the results of these models, precise control of energy administration can be achieved, leading to the most accurate application benefits without any inconvenience and ensuring efficiency and safety. 

While these electrical models correspond to a specific active medical device, they could also find application in other industrial contexts based on radiofrequency signals and the assembly of similar materials. This assumption arises from the requirement for active medical devices to implement sensors and deliver adequate energy to patients in order to provide efficient and safe therapies. 

The choice on behalf of adjusting by fractional order offers significant benefits for advancing low-cost instrumentation in industrial and medical research and warrants continued investigation.

## Figures and Tables

**Figure 1 sensors-24-02563-f001:**
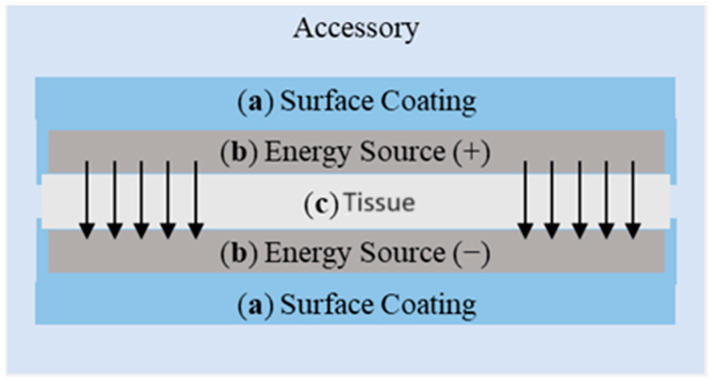
Arrangement of the different elements that comprise the accessory: (**a**) Surface coating, (**b**) energy source, (**c**) tissue. Notice that the arrows indicate the direction of the radiofrequency signal.

**Figure 2 sensors-24-02563-f002:**
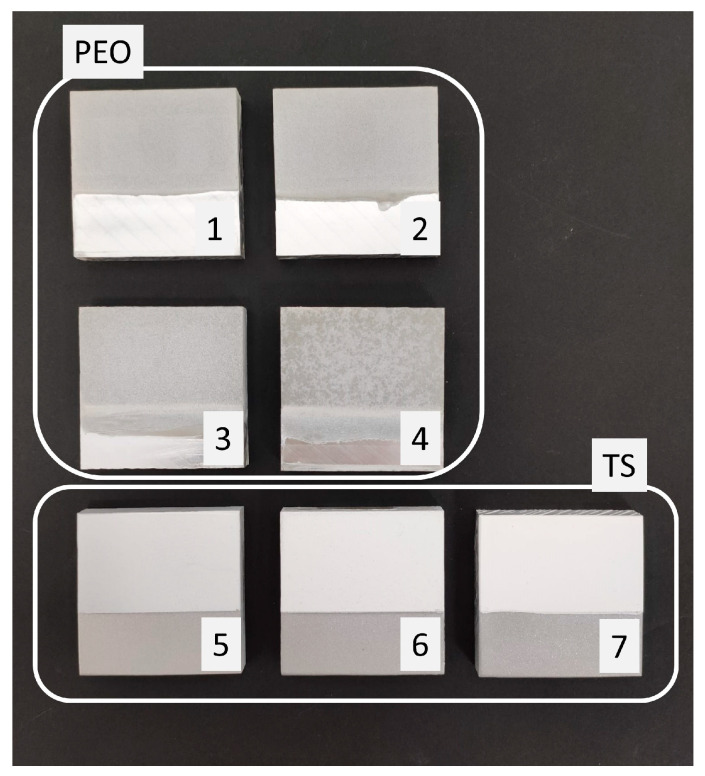
Aluminum samples with aluminum surface treated by PEO (Samples 1-2-3-4) and TS (Samples 5-6-7) with distinct thicknesses. Each sample corresponds to (1) 20 µm PEO, (2) 40 µm PEO, (3) 80 µm PEO, (4) 100 µm PEO, (5) 40 µm TS and 60 µm ceramic paint, (6) 70 µm TS and 100 µm ceramic paint, and (7) 250 µm TS.

**Figure 3 sensors-24-02563-f003:**
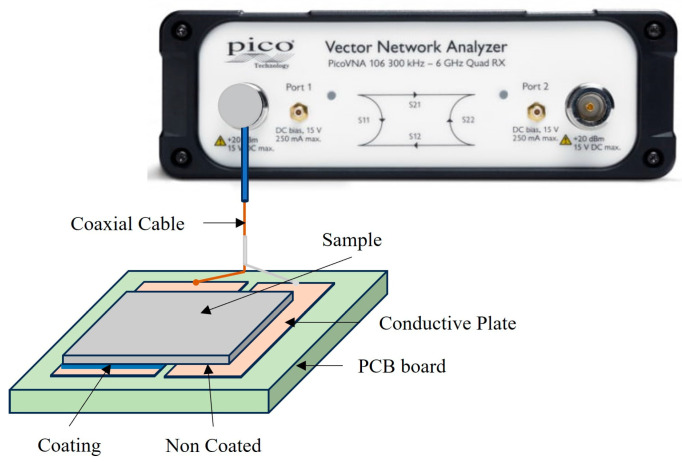
Schematic diagram of the measurement system based on vector network analyzer (VNA) connected to the load (coated sample in that work).

**Figure 4 sensors-24-02563-f004:**
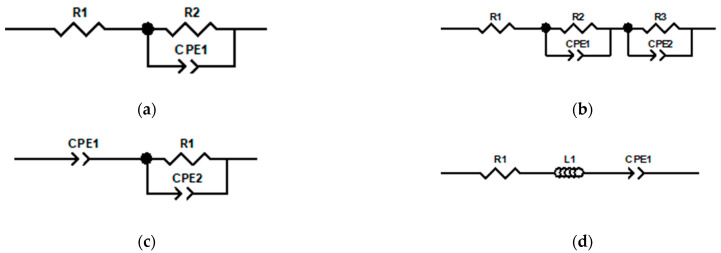
Equivalent electrical model of Cole family. (**a**) Single Dispersion Cole Model, (**b**) Double Dispersion Cole Model, (**c**) Capacitive Model, and (**d**) Inductive Model.

**Figure 5 sensors-24-02563-f005:**
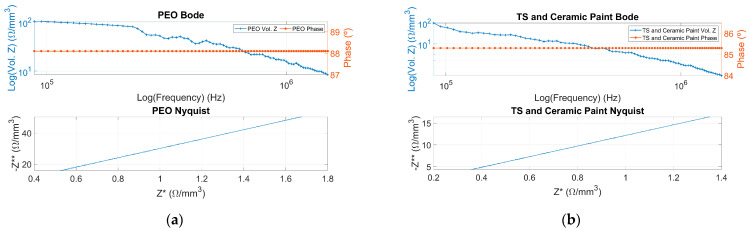
Bode and Nyquist plots. Top; volumetric impedance in logarithmic scale (**left axis**) and Phase (**right axis**) versus frequency in logarithmic scale from 0.5 MHz to 1 MHz. Bottom; negative imaginary part of impedance (Z**) versus real part of impedance (Z*). (**a**) Plasma electro-oxidation (PEO) coating, (**b**) thermal spray (TS) and ceramic paint coating.

**Figure 6 sensors-24-02563-f006:**
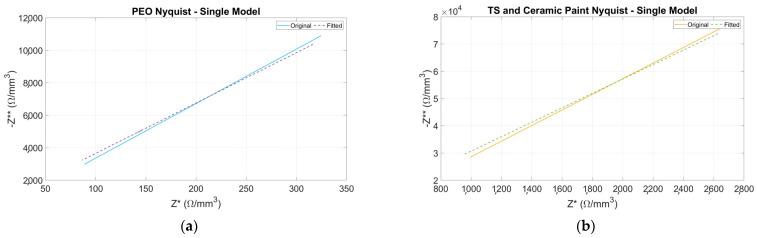
Nyquist plots. Negative imaginary part of impedance (Z**) versus real part of impedance (Z*). Solid line represents the original data, and dotted line represents the fitted data for single dispersion equivalent electrical. (**a**) PEO, (**b**) TS and ceramic paint.

**Figure 7 sensors-24-02563-f007:**
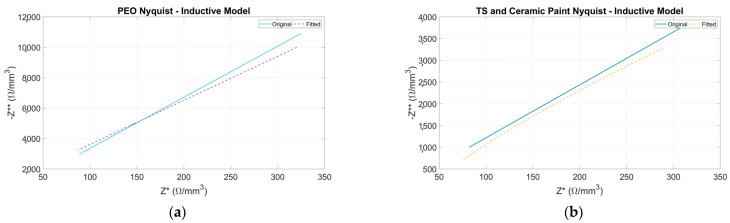
Nyquist plots. Negative imaginary part of impedance (Z**) versus real part of impedance (Z*). The solid line represents the original data, and the dotted line represents the fitted data for inductive equivalent electrical. (**a**) PEO, (**b**) TS and ceramic paint.

**Figure 8 sensors-24-02563-f008:**
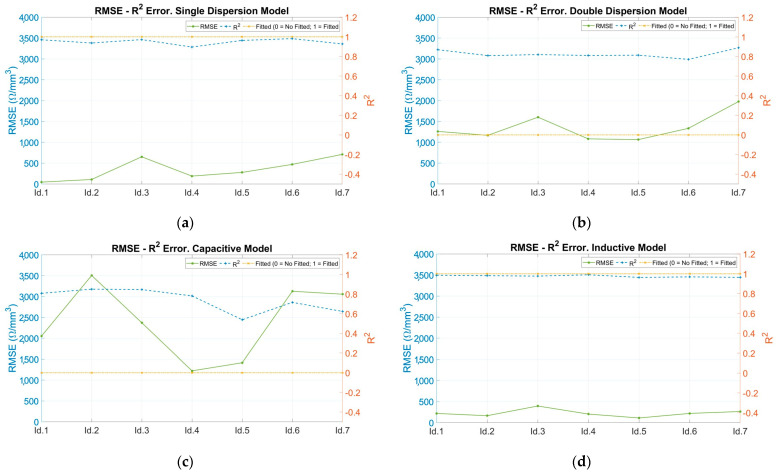
RMSE and R^2^ error in fitting models. RMSE (left axis and solid line), R2 (right axis and dotted line) and status of fitted model (right axis and dotted and dashed line) versus coating samples: 20 µm PEO (Id.1), 40 µm PEO (Id.2), 80 µm PEO (Id.3), 100 µm PEO (Id.4), 40 µm TS and 60 µm ceramic paint (Id.5), 70 µm TS and 100 µm ceramic paint (Id.6), and 250 µm TS (Id.7). (**a**) Single-dispersion model, (**b**) double-dispersion model, (**c**) capacitive model, and (**d**) inductive model.

**Figure 9 sensors-24-02563-f009:**
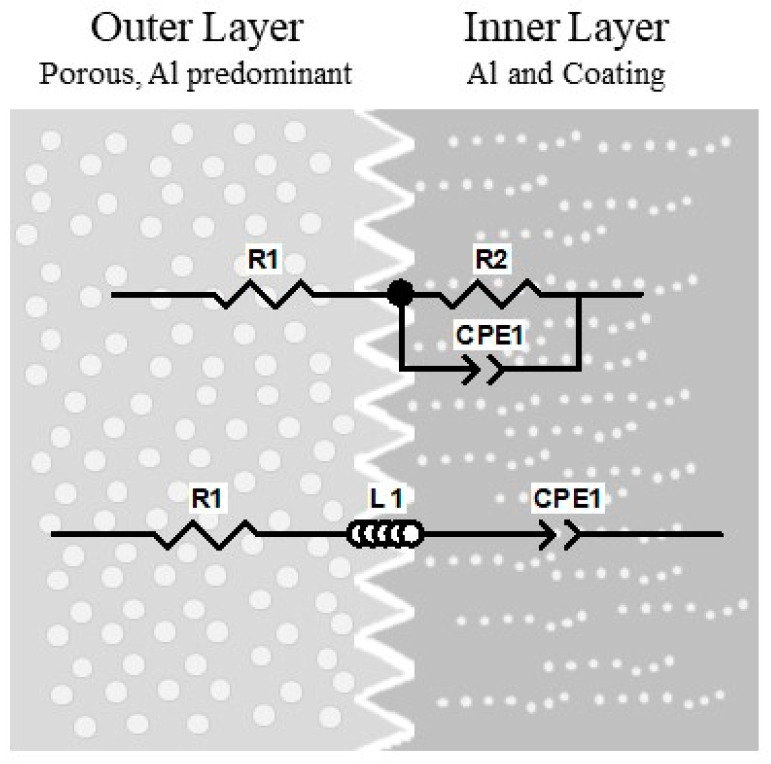
Physical coating structure (outer layer and inner layer) according to components of electrical equivalent.

**Table 1 sensors-24-02563-t001:** Surface coating configurations. PEO is plasma electro-oxidation; TS is thermal spray and CP is ceramic paint. Sample ID is the proposed identification of the coated samples during the work.

Sample ID	Surface Coating Method	Thickness (µm)
1	PEO	20
2	PEO	40
3	PEO	80
4	PEO	100
5	TS and CP	40 and 60
6	TS and CP	70 and 100
7	TS	250

**Table 2 sensors-24-02563-t002:** Approach of operational conditions of plasma electrolytic oxidation of aluminum alloys [32].

Parameters	Description
Substrate	Al
Applied current density (mA·cm^−2^)	100 (50 Hz)
Cell Voltage (V)	400 to 600
Duration (min)	90 to 150
Temperature (K)	343 to 353
Electrolyte	2 to 10 g dm^−3^ Na_2_SiO_3_
Compounds present in oxide layer	1 to 2 g dm^−3^ KOH
Substrate	α, γ-Al_2_O_3_

**Table 3 sensors-24-02563-t003:** Electrical parameter values adjusted to single-dispersion model for each coating sample. (Id.1 refers to 20 µm PEO; Id.2 refers to 40 µm PEO; Id.3 refers to 80 µm PEO; Id.4 refers to 100 µm PEO; Id.5 refers to 40 µm TS and 60 µm CP; Id.6 refers to 70 µm TS and 100 µm CP; Id.7 refers to 250 µm TS).

Circuit Element Values	Id.1	Id.2	Id.3	Id.4	Id.5	Id.6	Id.7
R1 (kΩ)	0.85	1.22	0.92	1.86	0.60	0.94	9.14
R2 (kΩ)	16.07	16.97	22.30	15.15	4.43	12.40	199.00
CPE1-T·10^−12^ (s^CPE-P^/Ω)	1.50	0.47	1.60	0.23	0.39	0.69	0.14
CPE1-P	0.81	0.99	0.74	0.83	0.94	0.99	0.78

**Table 4 sensors-24-02563-t004:** Electrical parameter values adjusted to inductive model for each coating sample. (Id.1 refers to 20 µm PEO; Id.2 refers to 40 µm PEO; Id.3 refers to 80 µm PEO; Id.4 refers to 100 µm PEO; Id.5 refers to 40 µm TS and 60 µm CP; Id.6 refers to 70 µm TS and 100 µm CP; Id.7 refers to 250 µm TS).

Circuit Element Values	Id.1	Id.2	Id.3	Id.4	Id.5	Id.6	Id.7
R1 (Ω)	90.42	60.95	131.10	462.60	148.60	62.20	182.10
L1 (mH)	0.02	0.42	0.65	0.32	0.25	0.49	4.60
CPE1-T·10^−12^ (s^CPE-P^/Ω)	129.00	47.30	58.90	193.00	289.00	84.10	6.09
CPE1-P	0.95	0.99	0.97	0.89	0.93	0.97	0.98

## Data Availability

The raw data supporting the conclusions of this article will be made available by the authors on request.

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
