# Peer review of "Fractional-Order Electrical Modeling of Aluminum Coated via Plasma Electro-Oxidation and Thermal Spray Methods to Optimize Radiofrequency Medical Devices"

_sensors, 2024, doi:10.3390/s24082563_

Round 1

Reviewer 1 Report

Comments and Suggestions for Authors

The authors of the article "Fractional-Order Electrical Equivalent of Aluminum Coated by Plasma Electro-Oxidation and Thermal Spray Methods in Radiofrequency Medical Devices" have made a series of samples fabricated by Aluminum Coated by Plasma Electro-Oxidation and Thermal Spray Methods as actuators in medical applications intended for social status. This is not directly related to the concept of the sensor as an area of scientific investigation. The work is interesting with an obvious addressability to very knowledgeable readers in the field. I suggest to the authors a rigorous scientific approach for a better understanding of the subject:

1. The abstract is atypical, practically reading it can significantly reduce the interest in reading the paper. This is due to an approach focused on the references (there is no need for references) and not on the subject of the study, i.e. novelty versus performance. I recommend to the authors an approach in line with the MDPI template;

2.  I consider that the Introduction is excessive in a review specific approach;

3. I recommend not only the presentation of the samples under investigation but especially the technology used to produce them. Also, here Figure 1, 2 does not respect the MDPI template, respecting the template would bring a plus, I do not recommend originality which does not bring anything good;

4. The experimental setup is presented very briefly, it is unclear what the "Error box" in Figure 3 refers to, probably it is the VNA which of course has a signal generator. I think that the type of VNA and its adaptation to the DUT is a plus being of great interest for an experimental validation of the measurements made, photography a plus;

5. The electrical model of a system/material should not be approached by justifying elements that have to do with the complexity of the circuit. The fact that a integer-order model implies a complex circuit does not justify the use of a fractional-order model. As a rule, we use a fractional-order electrical model when no matter how much we complicate an integer-order model we do not have a perfect fit with the experimental data. The fractional order model contains fewer components but it should not be forgotten that although it has fewer components it is from a mathematical perspective much more complex due to the fractionality of the components from the model;

6. Equation (1), the only equation in article, is correct but I think it needs to be justified and discussed in the context of a fractional order electrical model and concept;

7. Again, Figure 4 does not respect the template and is very low resolution;

8. The use of ZVIEW software provides the consistency needed for electrical modeling but is not a scientific contribution. For an addition here I recommend describing at length the constraints applied i.e. the initialization data and their motivation;

9. Figures 5, 6, 7 are part of a well-known logic, nor could they be otherwise, common practice;

10. In the Results and Discussion section there is an excess of Figures, without following the template. Resolution of these is very low, atypical notations, few experimental data points. Since they are tables of sample properties, I suggest reducing the number of Figures to a few representative examples;

11. Conclusions are in excess. The whole article needs to be revised, there are many sentences that repeat themselves or say the same thing in other words. It creates the impression that the authors still felt the need to say something more when there was nothing left to say.

Author Response

Dear Editor and Reviewers,

We would like to express our gratitude for the valuable and favorable feedback, comments, and discussion provided regarding the paper titled “Fractional-Order Electrical Modeling of Aluminum Coated via Plasma Electro-Oxidation and Thermal Spray Methods To Optimize Radiofrequency Medical Devices” that was submitted for publication in  Sensors.

The revised version of the manuscript, accompanying this letter, has undergone comprehensive editing, addressing a few misspellings and stylistic errors identified during the revision process.

Additionally, we have carefully addressed the comments provided by you, which are detailed in the attached document.

Reviewer 2 Report

Comments and Suggestions for Authors

This work is devoted to the study of the electrical response of aluminum-based materials with fractional order using two different spraying methods. It is worth noting that, in general, the ideology of the research and the task set in the work are not new. At the moment, there are many works in the literature devoted to this topic. This also reflects the amount of literature cited in the work. Thus, unfortunately, the manuscript does not provide a clear statement of purpose (with the exception of "expanding the electrical response. . ."), as well as justifications for the choice of research objects. The approaches used and the results so provided are very routine. Most of the illustrative information can be transferred to the Supplementary Materials for monotony and uselessness, and all data on impedance should be summarized in a single table. The work needs a significant rethink and restructuring in terms of its presentation. At the moment, I would like to make a recommendation on processing the article and sending the manuscript to a more specialized journal dedicated to obtaining materials with useful applied properties or applications for medicine (MDPI Materials, MDPI Journal of Functional Biomaterials, etc) . This is explained by the fact that from the point of view of the studied electrical responses as sensory properties, the work is quite routine and does not introduce significant novelty in this area.

Author Response

(The authors gave the same response as above.)

Round 2

Reviewer 1 Report

Comments and Suggestions for Authors

The authors of the revised version of the paper "Fractional-Order Electrical Equivalent of Aluminum Coated by Plasma Electro-Oxidation and Thermal Spray Methods in Radiofrequency Medical Devices." have succeeded in adding value to the article proposed for publication compared to the previous version. A first added value is the use of the MDPI template. Also on this occasion, the concepts are much better explained and the advantages of the approach are well highlighted by highlighting some original contributions even if the work is purely experiential by using professional equipment or software (available for direct use). The article has been revised in line with previous comments. In this form, I consider the presented study to be interesting and acceptable for publication. 

Reviewer 2 Report

Comments and Suggestions for Authors

After reviewing the work "Fractional-Order Electrical Modeling of Aluminum Coated via Plasma Electro-Oxidation and Thermal Spray Methods to Optimize Radiofrequency Medical Devices", the impression of the work has greatly improved. The authors removed some of the uninformative information, leaving only the necessary data, specified the approaches and methods used in the work, and also made significant changes in the introduction and expanded the conclusions obtained. At the moment, the work is at the good level and gives the reader an understanding of the need for research on plasma electro-oxidation and thermal spray methods of analysis and modification. In general, there are no more serious comments about the work. The manuscript can be published in present form.